# Effect of Siblings and Type of Delivery on the Development of Motor Skills in the First 48 Months of Life

**DOI:** 10.3390/ijerph17113864

**Published:** 2020-05-29

**Authors:** Miguel Rebelo, João Serrano, Pedro Duarte-Mendes, Rui Paulo, Daniel A. Marinho

**Affiliations:** 1Department of Sport Sciences, University of Beira Interior, 6201-001 Covilhã, Portugal; d.marinho@gmail.com; 2Department of Sports and Well-being Polytechnic Institute of Castelo Branco, 6000-266 Castelo Branco, Portugal; j.serrano@ipcb.pt (J.S.); pedromendes@ipcb.pt (P.D.-M.); ruipaulo@ipcb.pt (R.P.); 3Sport, Health & Exercise Research Unit (SHERU), Polytechnic Institute of Castelo Branco, 6000-266 Castelo Branco, Portugal; 4Research Centre in Sport Sciences, Health Sciences and Human Development (CIDESD), 6201-001 Covilhã, Portugal

**Keywords:** children, motor competence, motor development, type of delivery, presence of siblings, PDMS-2

## Abstract

This study aimed to verify whether the presence of siblings and the type of delivery had an influence on the motor skills development of children in the first 48 months of life. We developed a quantitative study with a sample of 405 children of both genders, divided according to the studied variables: children with siblings, children without siblings, children born via eutocic delivery, and children born via dystocic delivery. The instrument used in the study was the Peabody Developmental Motor Scales-2. Overall, the results indicated that children who had siblings had, on average, better outcomes regarding all motor skills (global and fine). Furthermore, those born via eutocic delivery, on average, had better outcomes regarding all motor skills (global and fine) when compared to children born via dystocic delivery. Thus, the presence of siblings in the family context and the type of delivery positively influenced motor development, especially after 24 months of age, showing that the presence of siblings providing cooperative activities through play and challenges improved cognitive, social, emotional, and physical development. Furthermore, a eutocic delivery, in addition to providing a better recovery from labor and the immediate affective bond between mother and child, also led to better results in terms of global and fine motor skills.

## 1. Introduction

Motor development is the process of changes in motor behavior, which not only involves the maturation of the central nervous system but also the interaction with the environment and the stimuli offered to the individual during their development [1]. The authors add that the two-way relationship between the individual and the environment is of marked importance and the transformations occur in a gradual and orderly manner.

Understanding, evaluating, and distinguishing motor development has been one of the main goals of many researchers investigating this area of knowledge. The attention given to the impact of a large set of variables regarding motor development rather than commonly studying motor development in isolation [1,2,3,4] has also been important, particularly regarding future research on child development in the first year of life.

According to Burns and Macdonald [5], researching human development means knowing the common characteristics of each age group, knowing how to recognize their individualities, and observing and interpreting behavior based on factors that influence human development. Among these factors, the authors emphasize heredity, genetic load (which establishes the potential of the individual, which may or may not develop), and the surrounding environment (which presupposes a set of environmental influences and stimulations that are capable of altering an individual’s behavioral patterns).

According to Santos, Dantas, and Oliveira [6], motor development in the early years of life is characterized by the acquisition of a wide repertoire of motor skills, which allows a child a complete mastery of their body in different postures, moving around the environment in various forms (walking, running, jumping, etc.), and manipulating various objects and instruments (receiving a ball, throwing a stone, kicking, writing, etc.).

According to Sugden and Wade [7], when children start movements, namely reflexes, during the first year of life, their arms and legs move apparently at random, but with the necessary precision to control posture, locomotion, and manipulation, we now know that these children’s “random” activities represent activity that is directly related to postural development and locomotion. At 24 months, children develop sufficient postural control to cope with many basic postural adjustments, such as being able to walk, explore, grab, and manipulate objects of various shapes and sizes. At 24 months, they are still unable to cope well with automatic and rapid movements relative to objects and other people in motion; as such, they require the assistance of others for a variety of motor activities. However, by 36 months, the child will already be able to walk, run, grab, and manipulate objects to a certain extent but the most advanced global motor skills and fine motor skills are still lacking [7].

Possible risk factors that influence children’s behavioral acquisitions have been the subject of several studies [3,8,9,10]. Motor skills are key and represent important milestones in a child’s motor development since they help them interact with objects and other people [11].

As such, it is important to understand how some factors influence the development of motor skills during childhood. Among these factors, the presence of older siblings has been highlighted in investigations since sibling relationships provide a basis for learning and socialization opportunities in several contexts [12]. In terms of motor development, older siblings can provide good role models that younger children can imitate [13,14,15,16], contributing to a decreased time needed by parents to teach basic motor skills, such as sitting independently and learning to walk [11]. Few studies have investigated the effects of older siblings on younger siblings’ motor skills but these report that children have a habit of copying older sibling’s behavior [15]; for example, children are prone to exploring objects and the environment with the presence of an older brother [17].

On the other hand, much has been said in the last decade regarding the significant increase in the number of dystocic births, where “planned” births have become a priority for mothers, which are usually induced via cesarean section. Khalaf et al. [18] report that birth via cesarean section is associated with lower motor and cognitive development at 9 months; furthermore, Perez-Rios, Ramos-Valencia, and Ortiz [19] underline the importance of developing intervention programs that promote breastfeeding and provide care, especially for women undergoing dystocic deliveries. On the other hand, Khadem and Khadivzadeh [20] and Li et al. [21] found no significant differences in the intelligence coefficients (IQ) between eutocic and dystocic births. However, “eutocic birth is generally considered to be the most natural form of birth with the least complications and least severity for women and the fetus” [22].

According to Isayama and Gallardo [23], the influence of context on the motor performance of premature children is important since this is where the biggest changes and adaptations occur. The results of these investigations are crucial for assisting in the orientation of professionals involved in the motor learning teaching processes, allowing them to understand the limitations and interventions needed when in contact with children. In this sense, it is pertinent to study motor development at an early stage of child development, where the growth and development of the child in the first two and three years of life are extremely marked compared to other periods of their life [24]. The objective of this study was to verify whether the presence of siblings and the type of delivery influenced the motor skills development of children in the first 48 months of life. Therefore, our study is relevant since it analyzed and investigated an important and little-studied age group, as well as being helpful for health, sports, and physical activity professionals in terms of interventions in a professional context with children while taking into account their motor development. 

## 2. Method

### 2.1. Participants

In this cross-sectional study, which took place in nurseries and kindergartens, a total of 405 subjects of both genders with ages (29.64 ± 8.83 months) between 12 and 48 months (F = 206, 29.35 ± 8.94 months; M = 199, 29.94 ± 8.73 months) took part. These subjects were from the urban environment in the district of Castelo Branco, Portugal, and were not involved in any guided motor skills session. For a better analysis of each age range, the participants were divided into the following three groups based on their age: from 12 to 23 months (N = 107, age = 18.79 ± 3.73 months), from 24 to 35 months (N = 153, age = 28.07 ± 3.35 months), and from 36 to 48 months (N = 145, age = 39.31 ± 3.56 months). These same children were also grouped according to the following study variables: children with siblings (N = 199, age = 30.61 ± 8.78 months), children without siblings (N = 206, age = 28.71 ± 8.81 months), born via eutocic delivery (N = 208, age = 30.70 ± 8.67 months), and born via dystocic delivery (N = 197, age 28.53 ± 8.89 months).

Initially, contacts were established with the institutions and nursery schools, which had collaboration protocols for the possibility of applying research instruments to children.

The following exclusion criteria were considered:Children who had been diagnosed with learning disabilities and/or developmental disabilities.Children with some type of diagnosed disability.Children aged under 12 months and over 48 months.

### 2.2. Instruments

The Peabody Developmental Motor Scales—Second Edition (PDMS-2) [25] was the instrument used to collect the motor profile information from the children studied. The PDMS-2 is one of the most widely used instruments for motor assessment. The scales were reviewed by Saraiva and Rodrigues [26] for the Portuguese population and allow for the performance of fine and global motor skills of children from birth to 71 months to be evaluated.

PDMS-2 results are reported regarding three domains of motor behavior: the fine motor quotient (FMQ), the global motor quotient (GMQ), and the total motor quotient (TMQ), which results from the previous two. This measuring instrument presents the child’s overall motor profile, as well as the result of the motor subtests that make up the scale [25].

The items were summed regarding the performance for each of the tests and their value was located in the reference table for the respective age (these reference values are appropriate for the Portuguese population based on the validation performed by Saraiva and Rodrigues [26], hence resulting in a standardized value and a percentile value that can be compared between ages). Then, the sum of the standardized values of the grouped tests allowed for obtaining the TMQ, GMQ, and FMQ based on the consultation of an appropriate table. Subsequently, the standard values were converted into a qualitative classification with categories ranging from “very poor” to “very superior,” as presented in Table 1.

The scales are standardized for the child population and have a mean value of 10 points (± 3) for each test (postural skills, locomotion skills, object manipulation skills, fine manipulation skills, visuo-motor integration skills), and a mean value of 100 (± 15) for motor quotients (global motricity and fine motricity).

The results of each test can be expressed using five types of final score: gross score, score by equivalent age, standard score, percentiles, and the motor quotients. Although the results can be expressed using five different ways, for the comparison between variables, the authors only suggest using the standard score, and it is for this reason that we only base our results on this score.

To obtain information about the participants, a child characterization form was created, in which information on the type of delivery and the presence or not of siblings in family involvement was provided.

### 2.3. Procedures

After obtaining approval from the data collection institution, an informed consent form was sent and requests were made to fill in the child characterization form, which allowed us to select the subjects after taking into account the study exclusion requirements. All ethical principles, international norms, and standards regarding the Helsinki Declaration and the Convention on Human Rights and Biomedicine were followed, respected, and preserved [27]. This project was approved by the Ethics Committee of the institution where the authors carry out their research.

According to Folio and Fewell [25], examiners using PDMS-2 as an assessment tool should understand the general procedures for administering the test. Data were collected by a single researcher who specializes in motor development.

The PDMS-2 administration was undertaken with each child individually and applied for approximately 45 to 60 min in a room or a large space with stairs. The assessment site was previously prepared to provide an environment with as little stimulation and distraction as possible for the children at a time that respected the daycare routines (namely the moments set aside for meals, bathing, or sleeping). The evaluations, when interrupted, were completed within five days, as established by the scale authors.

To correctly administer the instrument, the following rules were followed: the instructions for the items were given three times to each child. After performing the exercises, the evaluator recorded only the best result of the item; the child began the test at a point on the scale established by his or her age (these points were empirically determined to allow the examiner to start the test on an item that 75% of the children in the normative sample of that age had passed) and continuing until the child had failed to perform three consecutive items. The result of each item was 0 to 2 (0 does not perform, 1 performs with difficulty, and 2 performs well, according to the criteria defined in the instrument application manual). After the evaluation, the sum of each item was calculated until the final result was established regarding the global, fine, and total motor skills (which is the sum of the global and fine skills). Subsequently, the value of the sum of the items in each of the subscales was located in an age reference table, where a standardized value (from 1 to 20) was obtained, which was converted into a qualitative classification with seven categories ranging from “very superior” to “very poor” [28].

### 2.4. Statistical Analysis

For data coding, we used the IBM Statistical Package for the Social Sciences (SPSS v.25.0, IBM, Chicago, Illinois, IL, USA). In the first analysis, the normality of the sample was verified by applying the Kolmogorov–Smirnov test. As we obtained a non-normal distribution (sig. < 0.05) for all variables studied, we utilized the Mann–Whitney U test for independent samples, which allowed us to evaluate differences between groups. The method of inferences based on the magnitude of the effects was also performed using the following scale (d Cohen): 0–0.2, trivial; 0.21–0.6, low; 0.61–1.2, moderate; 1.21–2.0, high; ≥2.0, very high [29].

## 3. Results

Table 2 presents a general characterization of the sample, showing the minimum and maximum values (standard score), as well as the mean and standard deviation of the motor skills assessed in each age range. In global terms, the postural skills (PS) had the highest average values across all age ranges and locomotion skills (LS) had the lowest average values, except in the 36-to-48-months age group, in which it was the object manipulation skills (OMS) that obtained the lowest values. Global motricity (GM) showed better results only in the 12-to-23-months age range, while in the other age ranges, fine motricity (FM) was always the one with the best results.

Note that as age increased, children obtained better results regarding motor skills, except for object manipulation skills (OMS), where there was a decrease in results as the child grew, with younger children showing better results than the older ones.

It should be noted that regarding locomotion skills (LS), in the age range of 12 to 23 months, children had standard scores (7.61) below the expected average range (values between 8 and 12), while in the remaining age ranges and motor skills, all were within the standards considered “average” for the respective ages.

Table 3 presents the results of the comparative analysis within each age range between children who had siblings and those who did not have siblings.

In the 12–23-months age range, we can see that there were significant differences only in fine motricity, with the group of children with siblings obtaining better results (*p* = 0.005; *η^2^ =* 0.070; effect size: low). It is important to emphasize regarding the qualitative analysis that although there were no statistically significant differences, the group that did not have siblings presented, on average, better results regarding postural skills (*η^2^ =* 0.022; effect size: low), object manipulation skills (*η^2^ =* 0.010; effect size: trivial), and global motricity (*η^2^ =* 0.011; effect size: low).

In the 24–35-months age range, there were significant differences showing that the group of children who had siblings displayed, on average, better results regarding postural skills (*p* = 0.001; *η^2^ =* 0.070; effect size: low), locomotion skills (*p* = 0.001; *η^2^ =* 0.066; effect size: low), object manipulation skills (*p* = 0.039; *η^2^ =* 0.027; effect size: low), visuo-motor integration skills (*p* < 0.001; *η^2^ =* 0.186; effect size: moderate), global motricity (*p* < 0.001; *η^2^ =* 0.088; effect size: moderate), and fine motricity (*p* = 0.001; *η^2^ =* 0.073; effect size: low), but not in fine manipulation skills (*p* = 0.101; *η^2^ =* 0.017; effect size: low).

In the 36–48-months age range, on average, the group of children with a sibling showed better results in all motor skills, but there were only significant differences in locomotion skills (*p* = 0.014; *η^2^ =* 0.038; effect size: low), object manipulation skills (*p* < 0.001; *η^2^ =* 0.113; effect size: moderate), visuo-motor integration skills (*p* = 0.033; *η^2^ =* 0.033; effect size: low), and global motricity (*p* = 0.023; *η^2^ =* 0.035; effect size: low).

Table 4 provides the results of the comparative analysis within each age range between the children born via eutocic delivery with those born via dystocic delivery.

In the 12–23-months age range, we can see that there were no significant differences in any of the motor skills. However, using a qualitative analysis, we found that although there were no significant differences, the group that was born via eutocic delivery presented, on average, better results regarding all motor skills, postural skills (*η^2^ =* 0.006; effect size: trivial), locomotion skills (*η^2^ =* 0.017; effect size: low), object manipulation skills (*η^2^ =* 0.010; effect size: low), fine manipulation skills (*η^2^ =* 0.011; effect size: low), visuo-motor integration skills (*p* = 0.287; *η^2^ =* 0.010; effect size: trivial), global motricity (*η^2^ =* 0.006; effect size: trivial), and fine motricity (*η^2^ =* 0.020; effect size: low).

In the 24–35-months age range, significant differences were seen, with the group of children born via eutocic delivery having, on average, better results regarding postural skills (*p* = 0.049; *η^2^ =* 0.024; effect size: low), locomotion skills (*p* = 0.045; *η^2^ =* 0.025; effect size: low), and global motricity (*p* = 0.005; *η^2^ =* 0.051; effect size: low). However, although there were no significant differences, using a qualitative analysis, we can see that children born via dystocic delivery presented better results regarding fine manipulation skills (*η^2^ =* 0.015; effect size: low) and fine motricity (*p* = 0.927; *η^2^ =* 0.000; effect size: trivial).

In the 36–48-months age range, the group of children who were born via eutocic delivery showed, on average, better results regarding all motor skills, but there were only significant differences regarding object manipulation skills (*p* = 0.027; *η^2^ =* 0.032; effect size: low), fine manipulation skills (*p* = 0.024; *η^2^ =* 0.034; effect size: low), visuo-motor integration skills (*p* = 0.009; *η^2^ =* 0.045; effect size: low), global motricity (*p* = 0.013; *η^2^ =* 0.042; effect size: low), and fine motricity (*p* = 0.004; *η^2^ =* 0.043; effect size: low).

## 4. Discussion

The objective of the present study was to verify whether the presence of siblings and the type of delivery influenced the development of global and fine motor skills in the first 48 months of life, taking into account various age ranges. The observed data allowed us to verify, in general terms, that the majority of the children presented average values that were considered normal for the age range, except for the children aged 12 to 23 months regarding locomotion skills, in which the average values were considered below normal. It was also found that, as the age range increased, children showed better results regarding global motor skills but the results were more significant regarding fine motor skills. However, it is worth noting the fact that regarding object manipulation skills, for both the presence of siblings and the type of delivery, the children in the youngest group (from 12 to 23 months) showed better results than the older children (from 24 to 48 months).These results confirmed those of Gaul and Issartel [30], who also found a higher performance in older children, mainly regarding fine motor skills.

As for the variables studied, they were shown to influence the development of motor skills in the groups of children with siblings and those born via eutocic delivery, where on average, children in these groups obtained better results regarding different motor skills and in different age groups.

As for the presence of siblings variable, we found that between 12 and 23 months, there were no major differences between children with or without siblings; these results were ambiguous and are consistent with the study by Halpern et al. [8], who investigated children within the first 12 months and found a 90% probability of suspected motor delay in children living with four or more siblings; They explained this result, stating that parents with many children tended to be less attentive and willing to play and thus allow the child to explore its potential.

However, as children grow, there is a tendency for more and greater differences, with children with siblings achieving better results in global and fine motor skills. Despite the few studies that investigate the importance and influence of the presence of siblings in the family environment, in these age groups, our results are in line with most investigations, stating that children are in the habit of copying their sibling’s behavior [15] and that children are more likely to explore objects and the environment with siblings [17]. These results make sense since siblings provoke a series of stimuli that defy natural development; it is also known that at this stage, “imitation” is considered the normal model for stimulating growth processes [13,15,16]. The study by Martins et al. [31] contradicts our results, where their study refers to children who live in homes with more than seven residents as a risk factor for motor development. However, in our investigation, such facts were not found since the maximum number of siblings in the studied sample was three, with a frequency of only 9.1%.

Regarding the type of delivery variable, it was also found that for children aged between 12 and 23 months, there were no differences between the two types of delivery. However, for older children, these differences became more pronounced, especially in the 26-to-48-months range where the greatest differences between the types of delivery were noted; the best results were found for children born via eutocic delivery in terms of both global and fine motor skills. These results are in line with those obtained by Rodrigues and Silva [32], who found that those born via cesarean section (dystocic delivery) have worse locomotion, manipulation, visual, speech and language skills and personal autonomy compared to those born through a eutocic delivery; McBride et al. [33] also concluded that children born via a dystocic delivery had worse balance performance, fine motor coordination, and visual acuity compared to those born via a eutocic delivery. There are few investigations on this variable, especially in these age groups, but even so, it is essential to mention that according to recent studies, the type of delivery is an option in 80% of cases, where the mother usually opts for a dystocic delivery (cesarean, induced, forceps, etc.); this is usually done to avoid pain and sacrifice, but according to medicine, can bring serious complications for the mother and the baby. This is particularly important in Portugal, which currently has one of the highest rates of dystocic delivery, specifically cesarean delivery, in Europe.

These results are very useful and these variables must be taken into account, especially by health, sports, and physical activity professionals, such that they can intervene later to alleviate differences between children in each age range, regardless of the type of delivery or the presence of siblings.

As for the limitations of the present study, we consider that the time spent in collecting data with children of this age range makes the whole process difficult, as well as the lack of studies in these age groups. In future studies, it is suggested that other populations be investigated. Furthermore, it is important to understand whether these results are due to only these two variables or whether other variables may influence the results obtained, e.g., family involvement, the environment, the type of breastfeeding, and the controlled practice of physical activity in schools.

## 5. Conclusions

Overall, we concluded that the presence of siblings in the family context positively influenced motor development by providing cooperative activities through play and challenges that improve cognitive, social, emotional, and physical development. We also concluded that eutocic delivery, in addition to a better recovery from delivery and the immediate affective bond between mother and child, also enhanced the results of global and fine motor skills. These results are in line with the internationally held belief that the rate of dystocic deliveries should be reduced and consequently avoided. In this sense, the present study corroborates the need to maintain and disseminate the measures implemented at national and international levels, and to create new ways of reducing the rate of unnecessary dystocic deliveries with a view toward the best interests of the child.

## Figures and Tables

**Table 1 ijerph-17-03864-t001:** The Peabody Developmental Motor Scales—Second Edition (PDMS-2) subtest standard score values with the associated classification/description.

Standard Scores	Classification/Description
17–20	Very Superior
15–16	Superior
13–14	Above Average
8–12	Average
6–7	Below Average
4–5	Poor
1–3	Very Poor

**Table 2 ijerph-17-03864-t002:** Descriptive statistics for the PDMS-2 results in each age-range group.

PDMS-2	12–23 Months (N = 107)			24–35 Months(N = 153)			36–48 Months(N = 145)		
	Min	Max	M ± SD	Min	Max	M ± SD	Min	Max	M ± SD
PS	8	14	10.99 ± 1.39	8	16	11.75 ± 1.57	8	17	12.53 ± 2.14
LS	6	12	7.61 ± 1.25	5	12	8.69 ± 1.69	7	12	9.14 ± 1.10
OMS	5	16	10.21 ± 2.48	5	12	8.90 ± 1.87	7	12	8.93 ± 1.57
FMS	4	14	9.48 ± 2.11	7	14	10.14 ± 7.70	7	16	12.28 ± 2.53
VMIS	5	15	9.50 ± 1.95	5	13	9.37 ± 2.14	8	16	11.17 ± 2.16
GM	85	124	97.12 ± 8.14	62	115	98.01 ± 11.02	85	119	101.81 ± 7.98
FM	66	118	95.88 ± 8.81	82	118	98.33 ± 8.70	88	133	110.39 ± 11.71

PS—Postural Skills, LS—Locomotion Skills, OMS—Object Manipulation Skills, FMS—Fine Manipulation Skills, VMIS—Visuo-Motor Integration Skills, GM—Global Motricity, FM—Fine Motricity, M—Mean, SD—Standard Deviation.

**Table 3 ijerph-17-03864-t003:** Differences regarding the sibling presence variable in the PDMS-2 for each age range.

Age Range	PDMS-2	Presence of Siblings	N	M ± SD	*p*	*η^2^*	Effect Size
12–23 months	Postural skills	Yes	46	10.61 ± 1.47	0.114	0.022	0.299
		No	61	11.28 ± 1.27			
	Locomotion skills	Yes	46	7.74 ± 0.93	0.180	0.016	0.253
		No	61	7.51 ± 1.45			
	Object manipulation skills	Yes	46	9.96 ± 2.04	0.291	0.010	0.203
		No	61	10.39 ± 2.76			
	Fine manipulation skills	Yes	46	9.54 ± 1.72	0.744	0.001	0.062
		No	61	9.43 ± 2.37			
	Visuo-motor integration skills	Yes	46	9.96 ± 1.75	0.181	0.016	0.255
		No	61	9.16 ± 2.04			
	Global motricity	Yes	46	96.11 ± 7.02	0.277	0.011	0.209
		No	61	97.89 ± 8.87			
	Fine motricity	Yes	46	98.50 ± 7.77	**0.005 ***	**0.070**	**0.549**
		No	61	93.90 ± 9.09			
24–35 months	Postural skills	Yes	74	12.11 ± 1.83	**0.001 ***	**0.070**	**0.548**
		No	79	11.42 ± 1.19			
	Locomotion skills	Yes	74	9.11 ± 1.52	**0.001 ***	**0.066**	**0.532**
		No	79	8.29 ± 1.74			
	Object manipulation skills	Yes	74	9.22 ± 1.60	**0.039 ***	**0.027**	**0.332**
		No	79	8.61 ± 2.06			
	Fine manipulation skills	Yes	74	9.91 ± 1.66	0.101	0.017	0.266
		No	79	10.37 ± 1.73			
	Visuo-motor integration skills	Yes	74	10.28 ± 1.98	**<0.001 ***	**0.186**	**0.957**
		No	79	8.51 ± 1.91			
	Global motricity	Yes	74	101.74 ± 8.21	**<0.001 ***	**0.088**	**0.621**
		No	79	94.62 ± 12.19			
	Fine motricity	Yes	74	100.57 ± 8.71	**0.001 ***	**0.073**	**0.560**
		No	79	96.24 ± 8.21			
36–48 months	Postural skills	Yes	79	12.70 ± 1.91	0.423	0.004	0.130
		No	66	12.33 ± 2.38			
	Locomotion skills	Yes	79	9.39 ± 1.08	**0.014 ***	**0.038**	**0.400**
		No	66	8.83 ± 1.06			
	Object manipulation skills	Yes	79	9.44 ± 1.62	**<0.001***	**0.113**	**0.713**
		No	66	8.32 ± 1.27			
	Fine manipulation skills	Yes	79	12.46 ± 1.92	0.816	0.000	0.038
		No	66	12.08 ± 3.10			
	Visuo-motorintegration skills	Yes	79	11.54 ± 2.31	**0.033 ***	**0.033**	**0.354**
		No	66	10.71 ± 1.89			
	Global motricity	Yes	79	103.41 ± 7.66	**0.023***	**0.035**	**0.381**
		No	66	99.89 ± 7.98			
	Fine motricity	Yes	79	112.00 ± 9.80	0.155	0.014	0.237
		No	66	108.45 ± 13.39			

* *p* ≤ 0.05 using the Mann–Whitney U test; significant *p*-values and their associated effects are given in bold. N—Number of Subjects; M—Mean; SD—Standard Deviation.

**Table 4 ijerph-17-03864-t004:** Differences between the type of delivery in the PDMS-2 for each age range.

Age Range	PDMS-2	Type of Delivery	N	M ± SD	*p*	*η^2^*	d Cohen
12–23 months	Postural skills	Eutocic	48	11.10 ± 1.29	0.412	0.006	0.154
		Dystocic	59	10.90 ± 1.47			
	Locomotion skills	Eutocic	48	7.73 ± 1.09	0.162	0.017	0.264
		Dystocic	59	7.51 ± 1.37			
	Object manipulation skills	Eutocic	48	10.38 ± 1.94	0.286	0.010	0.205
		Dystocic	59	10.07 ± 2.85			
	Fine manipulation skills	Eutocic	48	9.73 ± 2.01	0.261	0.011	0.223
		Dystocic	59	9.27 ± 2.18			
	Visuo-motor integration skills	Eutocic	48	9.71 ± 1.74	0.287	0.010	0.202
		Dystocic	59	9.34 ± 2.11			
	Global motricity	Eutocic	48	98.08 ± 5.75	0.412	0.006	0.158
		Dystocic	59	96.34 ± 9.63			
	Fine motricity	Eutocic	48	96.65 ± 9.47	0.141	0.020	0.284
		Dystocic	59	95.25 ± 8.27			
24–35 months	Postural skills	Eutocic	76	11.86 ± 1.63	**0.049 ***	**0.024**	**0.312**
		Dystocic	77	11.65 ± 1.51			
	Locomotion skills	Eutocic	76	8.97 ± 1.65	**0.045 ***	**0.025**	**0.318**
		Dystocic	77	8.40 ± 1.69			
	Object manipulation skills	Eutocic	76	9.18 ± 1.94	0.056	0.023	0.308
		Dystocic	77	8.62 ± 1.77			
	Fine manipulation skills	Eutocic	76	9.96 ± 1.83	0.117	0.015	0.249
		Dystocic	77	10.32 ± 1.56			
	Visuo-motor integration skills	Eutocic	76	9.43 ± 2.16	0.673	0.001	0.068
		Dystocic	77	9.30 ± 2.19			
	Global motricity	Eutocic	76	100.21 ± 10.39	**0.005 ***	**0.051**	**0.465**
		Dystocic	77	95.95 ± 11.27			
	Fine motricity	Eutocic	76	98.12 ± 9.29	0.927	0.000	0.015
		Dystocic	77	98.55 ± 8.13			
36–48 months	Postural skills	Eutocic	84	12.80 ± 1.02	0.059	0.024	0.310
		Dystocic	61	12.16 ± 2.52			
	Locomotion skills	Eutocic	84	9.30 ± 1.59	0.109	0.016	0.259
		Dystocic	61	8.92 ± 0.99			
	Object manipulation skills	Eutocic	84	9.21 ± 1.67	**0.027 ***	**0.032**	**0.363**
		Dystocic	61	8.54 ± 1.34			
	Fine manipulation skills	Eutocic	84	12.69 ± 2.34	**0.024 ***	**0.034**	**0.376**
		Dystocic	61	11.72 ± 2.68			
	Visuo-motorintegration skills	Eutocic	84	11.56 ± 2.21	**0.009 ***	**0.045**	**0.435**
		Dystocic	61	10.62 ± 1.99			
	Global motricity	Eutocic	84	103.24 ± 7.90	**0.013 ***	**0.042**	**0.417**
		Dystocic	61	99.84 ± 7.71			
	Fine motricity	Eutocic	84	112.79 ± 10.76	**0.004 ***	**0.043**	**0.425**
		Dystocic	61	107.08 ± 12.22			

* *p* ≤ 0.05 using the Mann–Whitney U test; significant *p*-values and their associated effects are given in bold. N—Number of subjects; M—Mean; SD—Standard Deviation.

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
