# Peer review of "Effect of Siblings and Type of Delivery on the Development of Motor Skills in the First 48 Months of Life"

_ijerph, 2020, doi:10.3390/ijerph17113864_

Round 1
Reviewer 1 Report
The paper presents a well made and extremely interesting study, from a theoretical point of view but also from the point of view of the practical implications it may have.
However, I suggest the authors to consider some relevant integrations in order to reach a higher potential of the work. Below are some suggestions, which I think are relevant and could help to further strengthen this work.
Title: I suggest to the authors to change the title in order to make it more effective from a conceptual point of view. The terms "Presence of siblings" and "Type of childbirth" sound to me like variable names. For the reader, it would be more effective to give a more attractive title in terms of the relevant content of the work.
Introduction: I suggest to expand and enhance the part of the introduction by adding a paragraph illustrating the motor development in the relevant age range and also the possible measures of motor development, describing the components of the motor development in early stages and how these variables (siblings and type of childbirth) may influence at different stages of development different components.
Results: this work is carried out on a large group of children, so the data content should be exploited in this sense. In the first place, it would be important to verify the effect of age on performance at the Peabody: developmental trajectories could be drawn, as the number of subjects allows it. In this way, by tracing the motor development in the various areas it is possible to observe to what extent some areas have a linear development trajectory and which do not.
I also suggest to comment, in the first part of the results, on the general trend of the participants, the percentage of children within average performance and those at risk.
It would be relevant to carry out further analysis, either by checking the effects of age, but also by by raising it as one of the relevant questions in the work. The age range is very wide (12-48 months) and the development of the child is drastic at this stage. Therefore it would be relevant to consider the age in the analyses carried out and to add some further analyses.
What are the sub-components of motor development correlating across each other? over and above age differences?
But the most crucial question, which could be verified through regressions, is which of the independent variables, presence of siblings and type of birth predict which type of motor sub-component? The authors, with such a large group, Found differences depending on the two variables, on all motor measures in the analyses presented. But in which are the differences greater? On which aspects of motor development are these variables more important? Is age a mediator? Does the effect of these variables change according to age? Is it more pronounced in the younger or older children? All these questions are relevant.
These are just a few ideas, but I think the authors need to look at some relevant aspects in more detail. Otherwise the work asks only superficial and easily answered questions. In this way the contribution of the work could be greatly increased.
Author Response
Response to Reviewer 1 Comments
Point 1: Title: I suggest to the authors to change the title in order to make it more effective from a conceptual point of view. The terms "Presence of siblings" and "Type of childbirth" sound to me like variable names. For the reader, it would be more effective to give a more attractive title in terms of the relevant content of the work.

Response 1: Thank you for the suggestion. We changed the title of the article, removed the quotes and standardized the term “Type of Delivery” throughout the article.
Action in the corrected version of the manuscript:
Title: “Effect of Siblings and Type of Delivery on the development of motor skills in the first 48 months of life”.
Point 2: Introduction: I suggest to expand and enhance the part of the introduction by adding a paragraph illustrating the motor development in the relevant age range and also the possible measures of motor development, describing the components of the motor development in early stages and how these variables (siblings and type of childbirth) may influence at different stages of development different components.
Response 2: Thank you for the suggestion. We appreciate the suggestions and in the introduction we added a paragraph, which more precisely specifies the ages of our age range.
Action in the corrected version of the manuscript:
Line 54: “According to Sugden and Wade (2016), when children start movements, namely reflexes, during the first year of life, their arms and legs move apparently at random, but with the necessary precision to control posture, locomotion and manipulation, we now know that these children's "random" activities represent activity directly related to postural development and locomotion. At 24 months, children develop sufficient postural control to cope with many basic postural adjustments, being able to walk, explore, grab and manipulate objects of various shapes and sizes. At 24 months they are still unable to cope well with automatic and rapid movements in relation to objects and other people in motion, as such they require the assistance of others for a variety of motor activities. However by 36 months the child will already be able to walk, run, grab and manipulate objects to a certain extent, but the most advanced global motor skills and fine motor skills still lack difficulty (Sugden and Wade, 2016).”
Point 3: Results: this work is carried out on a large group of children, so the data content should be exploited in this sense. In the first place, it would be important to verify the effect of age on performance at the Peabody: developmental trajectories could be drawn, as the number of subjects allows it. In this way, by tracing the motor development in the various areas it is possible to observe to what extent some areas have a linear development trajectory and which do not..
Response 3: Thank you for the suggestion. We thank and agree with all the proposals. Thus, for a better understanding, we divided the sample into three age range groups: 12-23 months, 24-35 months and 36-48 months. Thus, with this division, we can already, as suggested, verify the effects and differences in the age groups taking into account the variables.
After these changes, all results, discussion and conclusions were changed.
Action in the corrected version of the manuscript:
Line 191: New Results;
Line 259: New Discussion;
Point 4: I also suggest to comment, in the first part of the results, on the general trend of the participants, the percentage of children within average performance and those at risk.
Response 4: Thank you for the suggestion. We accept and agree with the proposal, having commented in the first part of the results, the general trend of the participants, the percentage of children with average performance and those who are at risk.
Action in the corrected version of the manuscript:
Line 191: “Table 2 presents a general characterization of the sample, exposing the minimum and maximum values (Standard Score), the mean and standard deviation of the motor skills assessed in each age range. In global terms, the Postural Skills (PS) apresentaram had the highest average values in all age range and Locomotion Skills (LS) had the lowest average values, except in the 36 to 48 months, In which it was the Object Manipulation Skills (OMS) that obtained the lowest value. Global Motricity (GM) shows only better results in the 12 to 23 month age range, while in others age range, Fine Motricity (FM) is always the one with the best results.
Note that as age increases, children obtain better results in motor skills, except in Object Manipulation Skills (OMS), where there is a decrease in results as the child grows, with younger children showing better results that the older ones.
It should be noted that in Locomotion Skills (LS), in the age range 12 to 23 months, children have standard score values below (7,61) of the expected average range (values between 8 and 12) for age, while in the remaining ages ranges and motor skills, all are within the standards considered "average" for age.”
Point 5: It would be relevant to carry out further analysis, either by checking the effects of age, but also by by raising it as one of the relevant questions in the work. The age range is very wide (12-48 months) and the development of the child is drastic at this stage. Therefore it would be relevant to consider the age in the analyses carried out and to add some further analyses. What are the sub-components of motor development correlating across each other? over and above age differences?
Response 5: Thanks for the suggestion. We appreciate and agree that the age group was very broad. Thus, for better understanding, we divided the sample into three age groups: 12-23 months, 24-35 months and 36-48 months. Thus, with this division, we can already, as suggested, verify the effects and differences in the age groups, considering the variables.
After these changes, all results, discussions and conclusions were changed.
Action in the corrected version of the manuscript:
Line 191: New Results;
Line 259: New Discussion;
Point 6: But the most crucial question, which could be verified through regressions, is which of the independent variables, presence of siblings and type of birth predict which type of motor sub-component? The authors, with such a large group, Found differences depending on the two variables, on all motor measures in the analyses presented. But in which are the differences greater? On which aspects of motor development are these variables more important? Is age a mediator? Does the effect of these variables change according to age? Is it more pronounced in the younger or older children? All these questions are relevant.
Response 6: Thanks for the suggestion. We appreciate and agree with all the proposals and questions. However, a regression is not possible because in our study we have two categorical variables and for this reason it is not possible to perform a linear regression. But to answer all the very pertinent questions asked, by dividing the sample by age groups, it has already allowed us to answer all questions:
Action in the corrected version of the manuscript:
Line 261: “The observed data allow us to verify, in general terms, that the majority of the children presented average values ​​considered normal for the age range, except for the children aged 12 aos 23 meses, in Locomotion Skills, in which average values were considered below normal. It was also found that, as the age range increases, children show better results in global motor skills, but more expressive in fine motor skills. However, it is worth noting the fact that in the Object Manipulation Skills, both the presence of siblings and the type of delivery, that the children in the younger group (from 12 to 23 months) show better results than the older ones (from 24 to 48 months), the results being more expressive in the group of children without siblings and with dystocic delivery.”
Line 274: “As for the variable presence of siblings, we found that between 12 and 23 months (...) However, as children grow, there is a tendency for more and greater differences, with children with siblings achieving better results in global and fine motor skills.”
Line 291: “Regarding the type of delivery variable, it was also found that during the 12 and 23 months, there are no differences between the two type of delivery. However, it is as children grow that these differences become more pronounced, especially in the 26 to 48 months, that the greatest differences between the type of delivery are noted, highlighting the best results for children born by eutocic delivery, both in global and fine motor skills.”
Reviewer 2 Report
The Article “Importance of “Presence of Siblings” and “Type of Childbirth” in Motor Development in Children 12 to 48 Months” is well and quite clearly written but it requires a number of changes before it will be published:
- I would suggest correcting the title of the article. I would suggest taking out quotation marks from the title. As well as “type of childbirth” term used in different places differently, e. g. in 28 and 204 lines “Type of 2 Childbirth” and in line 85 line – “type of delivery”.
- Not very clearly described instrument scales used in the research: data presented in the 1 table for standard scores values and classification does not correct in the text and in the table - “very week” category or classification values is missing.
- Not clear information presented in lines 90-91 as well as in 129-130.
- In my opinion, to estimate children motor performance and compare it between 12 and 48 months not very correct. In these 2 years period there is so big difference in children motor development.
- I think presenting data about gender differences in motor performance results research would make much more valuable as well as including other statistical methods, e.g. hierarchical multiple regression analysis or etc. In such analysis all the factors could be evaluated together.
- Information about non-normal distribution in table 2 for each variable I think it is not necessary – it would be enough to mention that once in the notes or even in a text. As well as the titles of each column presented not correctly.
Before publication, in my opinion, article has to be improved.
Author Response
Response to Reviewer 2 Comments
Point 1: I would suggest correcting the title of the article. I would suggest taking out quotation marks from the title. As well as “type of childbirth” term used in different places differently, e. g. in 28 and 204 lines “Type of 2 Childbirth” and in line 85 line – “type of delivery”.
Response 1:Thank you for the suggestion. We changed the title of the article, removed the quotes and standardized the term “Type of Delivery” throughout the article.
Action in the corrected version of the manuscript:
Title: “Effect of Siblings and type of Delivery on the development of motor skills in the first 48 months of life”.
Line 29, and throughout the document: the term has been replaced by "delivery".
Point 2: Not very clearly described instrument scales used in the research: data presented in the 1 table for standard scores values and classification does not correct in the text and in the table - “very week” category or classification values is missing.
Response 2: Thank you for the suggestion. We appreciate the call for attention to the errors, having been corrected according to the application manual of the PDMS-2 (Folio & Fewell, 2000).
Action in the corrected version of the manuscript:
Line 136: was replaced, "Weak" by "Poor" and "Good" by "Superior".
Line 141: changes were made to the terms already mentioned and we added the missing line "1-3, Very Poor".
Line 181: The classification terms were corrected: "Very Good" for "Very Superior", and "Very Weak" for "Very Poor".
Point 3: Not clear information presented in lines 90-91 as well as in 129-130.
Response 3: Thank you for the suggestion. We correct errors and improve information for better understanding.
Action in the corrected version of the manuscript:
Line 101: “In this cross-sectional study, which took place in nurseries and kindergartens,and consisted a total of 405 subjects of both genders with ages (29.64 ± 8.83 months) understood between 12 and 48 months (F = 206, 29.35 ± 8.94 months; M = 199, 29.94 ± 8.73 months) from the urban environment in the district of Castelo Branco, Portugal and who do not conduct any guided motor skills session. For a better analysis of each age range, the participants were divided into 3 groups, according to their age: from 12 to 23 months (N = 107 - 18.79 ± 3.73), from 24 to 35 months (N = 153 - 28.07 ± 3.35) and from 36 to 48 months (N = 145 - 39.31 ± 3.56). These same children were also grouped according to the study variables: children with siblings (N = 199 - 30.61 ± 8.78 months); children without siblings (N = 206 - 28.71 ± 8.81 months); born by eutocic delivery (N = 208 - 30.70 ± 8.67 months); and born by dystocic delivery (N = 197 - 28.53 ± 8.89 months).”
Line 144: “Although the results can be expressed by 5 types of punctuation, for the comparison between variables, the authors only suggest the analysis through the standart scores, and it is for this reason that we only base our results on this score.”
Point 4: In my opinion, to estimate children motor performance and compare it between 12 and 48 months not very correct. In these 2 years period there is so big difference in children motor development.
Response 4: Thanks for the sugestion. We appreciate and agree with all the proposals. Thus, for a better understanding, we divided the sample into three age range: 12-23 months, 24-35 months and 36-48 months. Thus, with this division, we can already, as suggested, analyze the differences between variables, taking into account the age range.
After these changes, all results, discussions and conclusions were changed.
Action in the corrected version of the manuscript:
Line 191: New Results;
Line 259: New Discussion;
Point 5: I think presenting data about gender differences in motor performance results research would make much more valuable as well as including other statistical methods, e.g. hierarchical multiple regression analysis or etc. In such analysis all the factors could be evaluated together.
Response 5: Thanks for the sugestion. We are grateful for the proposal, which, in our opinion, has all the sentiment. But as for the gender differences with this sample, it has already been carried out in another study. However, a regression is not possible, because in our study we have two categorical variables and, for this reason, it is not possible to perform a linear regression or hierarchical multiple regression analysis.
However, to answer the very pertinent suggestions made, dividing the sample by age groups, it has already allowed us to better understand the relationship between age and the study variables. Thus drawing other analyzes and conclusions.
Point 6: Information about non-normal distribution in table 2 for each variable I think it is not necessary – it would be enough to mention that once in the notes or even in a text. As well as the titles of each column presented not correctly.
Response 6: Thanks for the sugestion. We agreed with the suggestion, checked the tables better and removed the table from normality, mentioning only the text, quoted below.
Action in the corrected version of the manuscript:
Line 185: “As we obtained a non-normal distribution (sig. <0.05) for all variables studied, we resorted to the Mann-Whitney U test for independent samples, which allowed us to evaluate differences between groups.”
Round 2
Reviewer 1 Report
I believe that the authors have made all the relevant changes in the paper that have reinforced a lot the work. Therefore in my opinion and to the best of my knowledge this paper is worth of being published in the journal.